# Cellular pharmacokinetic of methotrexate and its modulation by folylpolyglutamate synthetase and γ-glutamyl hydrolase in tumor cells

Fang Tang[1], Le Zou[2], Jingyao Chen[3], Fanqi Meng[2]*

1 Department of Pharmacy, The Affiliated Nanhua Hospital, Hengyang Medical School, University of South China, Hengyang, Hunan, China, 2 Pharmacy Department, The Seventh Affiliated Hospital, Sun Yat-sen University, Shenzhen, Guangdong Province, China, 3 Digestive Medicine Center, The Seventh Affiliated Hospital, Sun Yat-sen University, Shenzhen, China

* 197471179@qq.com

**Data Availability Statement:** All relevant data are within the manuscript.

**Funding:** The author(s) received no specific funding for this work.

## Abstract

### Background and purpose

Clinical studies showed that prolonged infusion of methotrexate (MTX) leads to more severe adverse reactions than short infusion of MTX at the same dose. We hypothesized that it is the saturation of folate polyglutamate synthetase (FPGS) at high MTX concentration that limits the intracellular synthesis rate of methotrexate polyglutamate (MTX-PG). Due to a similar accumulation rate, a longer infusion duration may increase the concentration of MTX-PG and, result in more serious adverse reactions. In this study, we validated this hypothesis.

### Experimental approach

A549, BEL-7402 and MHCC97H cell lines were treated with MTX at gradient concentrations. Liquid chromatograph-mass spectrometer (UPLC-MS/MS) was used to quantify the intracellular concentration of MTX-PG and the abundance of FPGS and γ-glutamyl hydrolase (GGH). High quality data were used to fit the cell pharmacokinetic model.

### Key results

Both cell growth inhibition rate and intracellular MTX-PG concentration showed a nonlinear relationship with MTX concentration. The parameter Vmax in the model, which represents the synthesis rate of MTX-PG, showed a strong correlation with the abundance of intracellular FPGS.

### Conclusion and implications

According to the model fitting results, it was confirmed that the abundance of FPGS is a decisive factor limiting the synthesis rate of MTX-PG. The proposed hypothesis was verified in this study. In addition, based on the intracellular metabolism, a reasonable explanation

**Competing interests:** The authors have declared that no competing interests exist.

was provided for the correlation between the severity of adverse reactions of MTX and infusion time. This study provides a new strategy for the individualized treatment and prediction of efficacy/side effects of MTX.

## 1 Introduction

Methotrexate (MTX) is a folic acid analog with broad-spectrum antineoplastic activity, which is used in the treatment of various solid tumors [1]. High dose methotrexate (HD-MTX) can effectively prevent cell resistance and extramedullary recurrence. It also plays an important role in combined chemotherapy [2]. MTX and its intracellular active metabolite, methotrexate polyglutamate (MTX-PG), can competently inhibit intracellular dihydrofolate reductase, block folate metabolism, inhibit purine synthesis and promote of cancer cell apoptosis [3]. However, after tumor cell death, MTX causes side effects such as mucosal damage, myelosuppression, and liver and kidney damage [4, 5].

In clinical practice, it was observed that the severity of MTX toxicity is correlated with the infusion time. At the same dose, the side effects caused by slower infusion of MTX are more serious, and the regularity is contrary to the general rule of clinical medication [6].The range of HD-MTX infusion time is quite wide (2 to 48 hours). There is a lack of awareness abourt the effects of the infusion rate, and clinicians usually slow down the infusion rate to reduce side effects, which leads to more serious side effects [7]. Related studies [8] have shown that slow administration of MTX increases MTX-PG levels in the kidney and other tissues, enhancing intracellular oxidative stress and leading to tissue damage. The higher risk of side effects after slower infusion at the same dose may be related to the increased level of intracellular MTX-PG.

There is no obvious rate-limiting step in the cell entry of MTX. At low doses ($<20$ μM), MTX is actively transported into cells primarily by transporters such as the reducing folate carrier (RFC), the proton-coupled folate transporter (PCFT), and the organic anion transporter polypeptide (OATP). In the presence of high ($>20$ μM) concentrations of extracellular MTX, it enters the cells mainly through passive diffusion and is then metabolized into MTX-PG [3].

The concentration of intracellular MTX-PG is mainly affected by two metabolic enzymes, folate polyglutamate synthetase (FPGS) and γ-glutamyl hydrolase (GGH). FPGS and GGH are key enzymes for MTX/MTX-PG transformation and are also critically involved in the intracellular accumulation of MTX-PG [9–11] The intracellular prototype drug methotrexate (MTX-$PG_1$) binds to glutamate residues (generally 2–6) in the presence of FPGS, thereby producing MTX-PG (MTXPG$_{2-7}$). GGH catalyzes the hydrolysis of MTX-PG to MTX, which is then transported to the extracellular space. Compared with MTX, MTX-PG is not easily transported to the extracellular pace due to its larger chain length. Thus MTX-PG accumulates intracellularly and increases the risk of toxicity [12]. The difference in FPGS and GGH among different people is an important reason for individualized differences in the efficacy of MTX [13].

Although several studies reported thar slower MTX infusion leads to more serious side effects, the underlying mechanism is still unclear. Mikkelsen et al. [6] proposed that intracellular MTX-PG accumulates more during slower infusion at the same dose. H J Lenz et al. [14] found that in the CHO AUXB1 cell line, the gene expression level of FPGS was linearly associated with t MTX-PGs levels. Amy J. Galpin et al. [10] used the level of MTX-PG as a measure

of FPGS activity, and found that the expression level of FPGS mRNA in NALM6 cells was three times of that in CEM cells, and its activity was also 3–4 times higher in NALM6 cells. Changes in the efficacy and side effects of MTX may be determined by the expression of key proteins in its intracellular metabolic process.

We hypothesized thar FPGS as a key enzyme that determines the synthesis of intracellular MTX-PG, is easily saturated with high concentrations of MTX, resulting in a similar concentration of MTX-PG after adopting different infusion rates for the same dose. Therefore, the amount of intracellular MTX-PG depends largely on the exposure time of MTX. During slower infusion, the blood concentration of MTX is maintained at a high level for a longer duration, which increases the intracellular level of MTX-PG and the risk of serious side effects.

In this study, we studied the metabolism of MTX and measured the FPGS and GGH levels using the newly established liquid chromatograph-mass spectrometer (UPLC-MS/MS) absolute quantification method. A cellular pharmacokinetic model of MTX was established and the correlation between the abundance of key enzymes, FPGS and GGH, and corresponding functional parameters was investigated. We also elucidated the mechanism linking slower infusion to more side effects.

## 2 Materials and methods

### 2.1 Chemicals and reagents

Methotrexate (CAS: 59-05-2) was obtained from the National Institutes for Food and Drug Control (Beijing, China). Fetal bovine serum (FBS), Dulbecco's modified Eagles medium (DMEM), RPMI-1640, trypsin containing 0.25% EDTA, and phosphate-buffered saline (PBS) were purchased from Thermo Fisher Scientific (Waltham, MA, USA). The total protein extraction kit was supplied by Bestbio Co., LTD (Shanghai, China). Bradford protein quantitation assay kit was obtained from Beyotime Biotech Co., LTD (Nanjing, China). Trypsin Gold, Mass Spectrometry Grade was purchased from Promega Corporation (Madison, WI, USA). The synthetic peptide SGLQVEDLDR and YLESAGAR (≥99.5% purity) and the stable isotope-labeled internal standard (SIL-IS) SGL($^{13}$C,$^{15}$N)QVEDLDR and YLESAGAR($^{13}$C,$^{15}$N) (≥99.5% purity) were synthesized by Shanghai Science Peptide Biological Technology Co., LTD (Shanghai, China). HPLC-grade acetonitrile, formic acid and methanol were obtained from Merck KGaA (Darmstadt, HE, DE). Human serum albumin (HSA), ammonium bicarbonate (NH$_4$HCO$_3$), dithiothreitol (DTT) and trifluoroacetic acid (TFA) were purchased from Sigma-Aldrich (St. Louis, MO, USA). Water was purified and deionized with a Milli-Q Direct 8 system manufactured by Merck Millipore (Billerica, MA, USA).

### 2.2 Establishment and validation of a quantitative method for FPGS and GGH

A novel UPLC-MS/MS-based method was developed to accurately quantify the intracellular abundance of FPGS and GGH. The surrogate peptides generated by trypsin-mediated digestion of proteins were monitored as the targets for quantification [15–18]. SGLQVEDLDR and YLESAGAR were selected as the surrogate peptides of FPGS and GGH based on in silico prediction models and experimental data. Isotopically-labeled peptides, SGL($^{13}$C,$^{15}$N)QVEDLDR and YLESAGAR($^{13}$C,$^{15}$N), were synthesized as internal standards. The appropriate chromatographic conditions, mass-spectrometry parameters and the optimum process of sample digestion [19] were determined based on experimental data. Several key parameters were measured to validate the UPLC-MS/MS method. The detection linearity, lower limit of quantification (LLOQ), accuracy, precision, blank matrix effect and stability were assessed.

## 2.3 Cytotoxicity analysis of gradient concentration of MTX

The human lung cancer cell line A549, the human liver cancer cell line BEL-7402, and the human highly metastatic liver cancer cell line MHCC97H were purchased from the Xiangya Cell Bank of Central South University (Changsha, China). RPMI-1640 medium with 10% FBS was used for A549 and BEL-7402, and DMEM medium with 10% FBS was used for MHCC97H. Cell lines were cultured in a humidified atmosphere of 5% carbon dioxide at 37 ˚C.

Cell lines in the logarithmic growth phase (cell density $5 \times 10^4$) were incubated with MTX at a concentration of 0.01, 0.03, 0.1, 0.3, 1, 3, 10, and 30 μM. According to the literature [10], 24h was set as the end point of the cytotoxicity experiment, the absorbance of the cell solution (100 μL) was measured using the CCK-8 method, and the cell activity was measured. The effect of MTX concentration on the cell growth inhibition rate was analyzed. The intracellular concentration of MTX-PG was measured using the MTX-PG quantification method, The result is expressed as $pmol \times 10^6$. The effect of MTX concentration on the intracellular MTX-PG accumulation and the correlation between MTX-PG concentration and cell growth inhibition rate were analyzed.

## 2.4 Intracellular metabolism of MTX

A549, BEL-7402, and MHCC97H in logarithmic growth phase (cell density $1 \times 10^6$) were incubated with MTX at a concentration of 0.03, 0.1, 0.3, 1, and 3 μM. 2 mL of cell solution was obtained at 0.5, 1.5, 3, 6, 9, and 12 h, respectively. Intracellular MTX-PG was extracted by sonicating cells in phosphate buffer saline. Samples were stored at -80 ˚C until analysis. MTX and MTX-PG in cell samples and culture medium were extracted using the OASIS MAX solid-phase extraction column, and the concentration of MTX was measured using the UPLC-MS/MS method. Data on the intracellular concentration of MTX-PG were obtained after treatment with different concentrations of MTX at different time points.

## 2.5 Correlation between key enzyme level and intracellular metabolism of MTX

**2.5.1 Determination of enzyme level in different cell lines.** A549, BEL-7402, and MHCC97H in the logarithmic growth phase (cell density $1 \times 10^6$) were incubated with MTX at a concentration of 0.03, 0.1, 0.3, 1, and 3 μM. 2 mL of cell solution was obtained at 0.5, 1.5, 3, 6, 9, and 12 h, respectively. Total cellular protein was extracted and its concentration was determined using a Bradford protein quantitation assay kit.

The diluted protein solution was heated at 60 ˚C for 20 min, adding $DTT/NH_4HCO_3$ to break the disulfide bond and IAA to reduce the protein. Then the solution was dark incubated with mass spectrometry grade trypsin for 12 h. The reaction was ended using trifluoroacetic acid.

The abundance of FPGS and GGH in different cell lines was determined using the UPLC-MS/MS protein absolute quantification method. The effects of different concentrations of MTX on the expression of k FPGS and GGH were analyzed.

**2.5.2 Establishment of intracellular pharmacokinetic model.** Based on previous studies on the intracellular kinetics of MTX and MTX-PG, a kinetic model was established describing the transformation of the prototype drug MTX (MTX-PG$_1$) to its intracellular glutamate-linked metabolite (MTX-PG$_{2-7}$) (Fig 1).

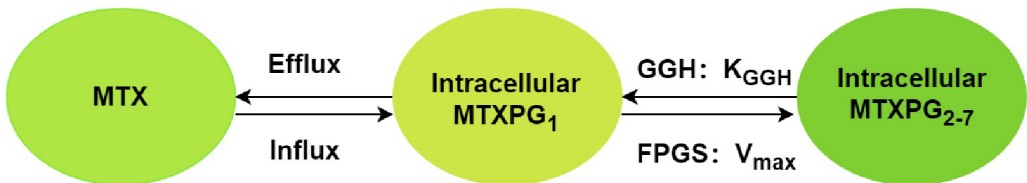

**Fig 1. Cellular and external MTX transport process and intracellular MTX metabolism process (Vmax is the saturation rate when intracellular MTX-PG$_1$ is converted to MTX-PG$_{2-7}$; $k_{GGH}$ is the kinetic constant).**

A kinetic model describing the transport and intracellular metabolism of the prototype drug MTX (MTX-PG$_1$) was established. The equation is as follows:

$$\frac{dy_2}{dt} = k_{in} \cdot y_1 - k_{out} \cdot y_2 - v_{max-FPGS} + k_{GGH} \cdot y_3$$

$$\frac{dy_3}{dt} = v_{max-FPGS} - k_{GGH} \cdot y_3$$

Parameter description: kin and kout represent the kinetic constants of MTX transferred into and out of cells, respectively, Vmax is the saturation rate when intracellular MTX-PG$_1$ is converted to MTX-PG$_{2-7}$; $k_{GGH}$ is the kinetic constant of hydrolysis process. $y_1$ represents the extracellular MTX concentration, $y_2$ is the intracellular MTX concentration, and $y_3$ represents the intracellular MTX-PG$_{2-7}$ concentration.

**2.5.3 Correlation analysis between functional parameters and protein abundance.** The initial values of the parameters kin, kout, $v_{max}$-FPGS and kGGH in the cell pharmacokinetic equation were set based on the literature. The experimentally measured MTX-PG, FPGS and GGH data were substituted into the model, and the least-square method was used to fit the parameters in the model.

Model fittings were performed on model parameters Vmax, $k_{GGH}$, FPGS, and GGH expression levels.

## 2.6 Statistical analysis

Statistical analysis was performed using SPSS for Windows version 22.0 (IBM Corporation, Armonk, NY, USA). Student's t-test and Pearson correlation analysis were used to assess the difference and correlation between two groups. P-value less than 0.05 was considered statistically significant. Model fitting was done using 1stOpt software (7D-Soft High Technology Inc., China).

# 3 Results and discussion

## 3.1 Validation of quantitative methods for FPGS and GGH

The established method quantified the abundance of FPGS (Y = 0.00194X-0.000237, $R^2$ = 0.9993, weight: 1/X$^2$) and GGH (Y = 0.000465X+0.000316, $R^2$ = 0.9992, weight: 1/X$^2$). The detection signals of both peptides (SGLQVEDLDR and YLESAGAR) were linear at the range of 0.1–100 ng/mL and the lowest limit of quantification (LLOQ) was 0.1 ng/mL. Since the FPGS and GGH depleted cell matrix was not available, 5% human serum albumin (HSA) in PBS was applied as a surrogate matrix in which all peptides showed good specificity. For both two peptides, the intra-day and inter-day precisions (%RSD) were within ±10% and accuracy

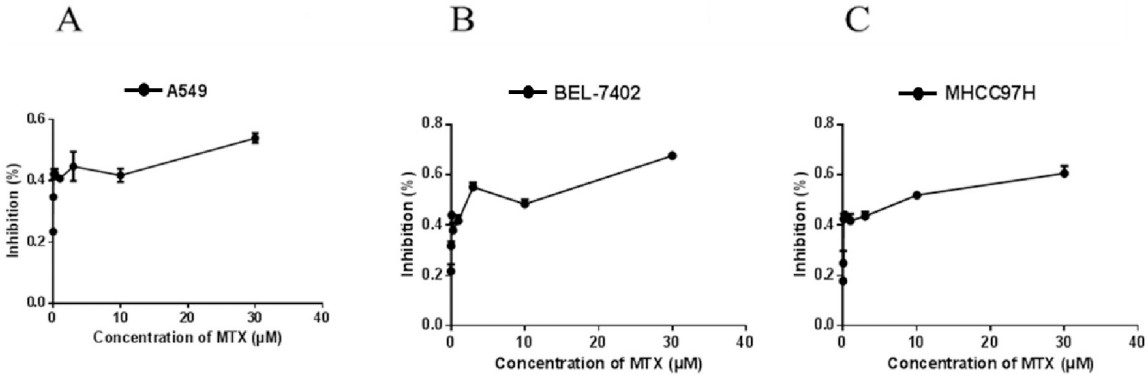

**Fig 2.** The proliferation inhibition rate curves of three kinds of cells treated with different concentrations of MTX for 24 h (A: the inhibition rate of A549 cell line treated with gradient MTX for 24 h; B: the inhibition rate of BEL-7402 cell line treated with gradient MTX for 24 h; C: the inhibition rate of MHCC97H cell line treated with gradient MTX for 24 h).

(% RE) was within ± 15%. All samples were stable under different conditions (24h at room temperature, 20d at -20˚C, and 3 freeze-thaw cycles at -20˚C).

UPLC-MS/MS is high-throughput, sensitive, accurate and precise, and has been successfully applied in detecting trace proteins in complex matrices. The diversity of intracellular proteins can affect the precision of results. The separation and purification of total cellular proteins by electrophoresis can improve the accuracy of quantitative results.

## 3.2 Correlation between MTX concentration and cell inhibition rate and MTX-PG accumulation

We measured the cell inhibition rate and intracellular metabolism of MTX to verify that the cytotoxicity and intracellular metabolism of MTX in vitro were consistent with clinical phenomena The inhibition rate and the intracellular MTX-PG accumulation curve of three cell lines were obtained after treatment with 0.03~30 μM MTX for 24h (Fig 2).

The intracellular concentrations of MTX-PG in A549, MHCC97H and BEL-7402 showed a similar trend under parallel treatment (0.03~30 μM MTX treatment for 24h)

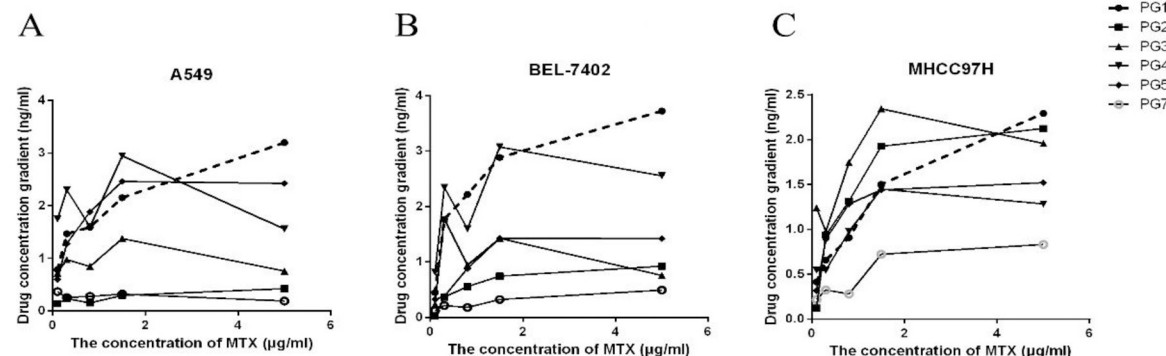

**Fig 3.** Intracellular MTX-PG concentration curve of different cell lines treated with gradient concentration of MTX for 24h (The dotted line represents the change trend of intracellular MTX-PG1; the solid line represents the change trend of MTX-PG2-7; A: the change of MTX-PGS concentration in A549 cells after 24 h of gradient MTX treatment; B: the change of MTX-PGS concentration in BEL-7402 cells after 24 h of gradient MTX treatment; C: the change of MTX-PGS concentration in MHCC77H cells after 24 h treatment with gradient MTX).

**Table 1. Pearson correlation coefficient between treatment concentration and metabolite concentration.**

|  | MTX-PG$_1$ | MTX-PG$_2$ | MTX-PG$_3$ | MTX-PG$_4$ | MTX-PG$_5$ | MTX-PG$_6$ | MTX-PG$_7$ |
|---|---|---|---|---|---|---|---|
| A549 | 0.946** | 0.308 | 0.107 | 0.270 | 0.483 | 0.318 | -0.775 |
| BEL-7402 | 0.813** | 0.427 | 0.142 | 0.487 | 0.558 | 0.874** | 0.591 |
| MHCC97H | 0.905** | 0.432 | 0.542 | 0.295 | 0.802** | 0.454 | 0.601* |

** indicates very strong correlation;

* indicates strong correlation

(Fig 3). MTX (MTX-PG$_1$) showed a dose-dependent increase with treatment methotrexate concentration. The intracellular concentration of MTX-PG$_{2-7}$ showed a trend similar to that of cell inhibition rate. After exceeding a certain threshold of treatment concentration, the concentration of MTX-PG did not significantly change as the concentration of MTX increased.

Person correlation analysis was performed on related variables to assess the effect of external drug concentration on intracellular metabolite concentration, Data is shown in the Table 1.

In all three cell lines, the accumulation concentration of intracellular MTX was strongly correlated with the concentration of MTX, which could be explained by the influx pathway of MTX. MTX can be actively transported by RFC, PCFT, OATP1B1 and other transporter proteins, or passively diffused into cells at high concentrations. Therefore, there is no obvious rate-limiting step in the influx of MTX. The concentration of intracellular MTX is directly affected by the concentration of the extracellular drug. However, it was found [20] that most of the intracellular concentrations of MTXPG$_{2-6}$ after 24 hours of gradient treatment had no correlation with the treatment dose. MTX is metabolized to more active MTX-PG in cells, and MTX-PG is subsequently cleaved in lysosomes by GGH. There was no significant correlation between the accumulation of intracellular MTX-PG$_{2-7}$ and the concentration of MTX in different cell lines. The intracellular concentration of MTX was positively correlated with the extracellular concentration of MTX, indicating that there was no rate-limiting step affecting the accumulation of MTX-PG. Intracellular MTX is catalyzed by FPGS to produce MTX-PG$_{2-7}$, and MTX-PG$_{2-7}$ is hydrolyzed to MTX by GGH to excrete cells. The intracellular concentration of MTX-PG$_{2-7}$ reaches a plateau in high concentrations of MTX, which may be related to the saturation of FPGS. The substrate of GGH is produced by FPGS, and the concentration is relatively low. It is hypothesized that there is no saturation in the hydrolysis process, which has little effect on the accumulation of intracellular MTX-PG$_{2-7}$.

Intracellular conversion of MTX to MTX-PG enhances its pharmacological action. According to the literature, MTX-PG$_{3-5}$ is the main factor affecting drug efficacy. The correlation between extracellular MTX concentration, intracellular MTX-PG concentration and the cell inhibition rate was analyzed to explore why the gradual stabilization of the cell inhibition rate is curve-shaped.

According to the results(Table 2), the intracellular concentration of MTX was poorly correlated with the cell inhibition rate. The intracellular concentration of MTX-PG$_{2-7}$ was highly correlated with the cell inhibition rate, and both increased first and then they reached a plateau with further increase in the treatment dose of MTX. It is suggested that the limited formation rate of MTX-PG$_{2-7}$ prevents further increase in cytotoxicity after exposure to high concentration of MTX.

**Table 2. Pearson correlation coefficient between extracellular methotrexate concentration, intracellular MTX-PG concentration and the cell inhibition rate.**

|  | MTX | MTX-PG$_1$ | MTX-PG$_{2-7}$ |
|---|---|---|---|
| A549 | 0.423 | 0.416 | 0.859 |
| BEL-7402 | 0.369 | 0.382 | 0.820 |
| MHCC97H | 0.458 | 0.426 | 0.837 |

No correlation was defined as 0<R<0.2, weak correlation was defined as 0.2<R<0.4, moderate correlation was defined as 0.4<R<0.6, strong correlation was defined as 0.6<R<0.8, and extremely strong correlation was defined as 0.8 < R < 1.0.

## 3.3 Changes in intracellular and extracellular concentrations of MTXPG in different cells treated with different concentrations of MTX

The concentration of intracellular MTX-PG may be affected by various factors, including transporter expression, metabolic enzyme activity, and extracellular MTX concentrations. In this study, UPLC-MS/MS was used to obtain high-quality data on the intracellular concentration of MTX-PG under different conditions, which was used to fit the cytopharmacokinetics model of MTX (Figs 4 and 5).

At different time points, the concentration of extracellular MTX remained constant, whereas the concentration of MTX-PG$_{2-7}$ was extremely low and undetectable. The results demonstrated that MTX-PG$_{2-7}$ is difficult to efflux and has a long intracellular retention time. There are differences in the accumulation process of MTX-PG between different cell lines, and

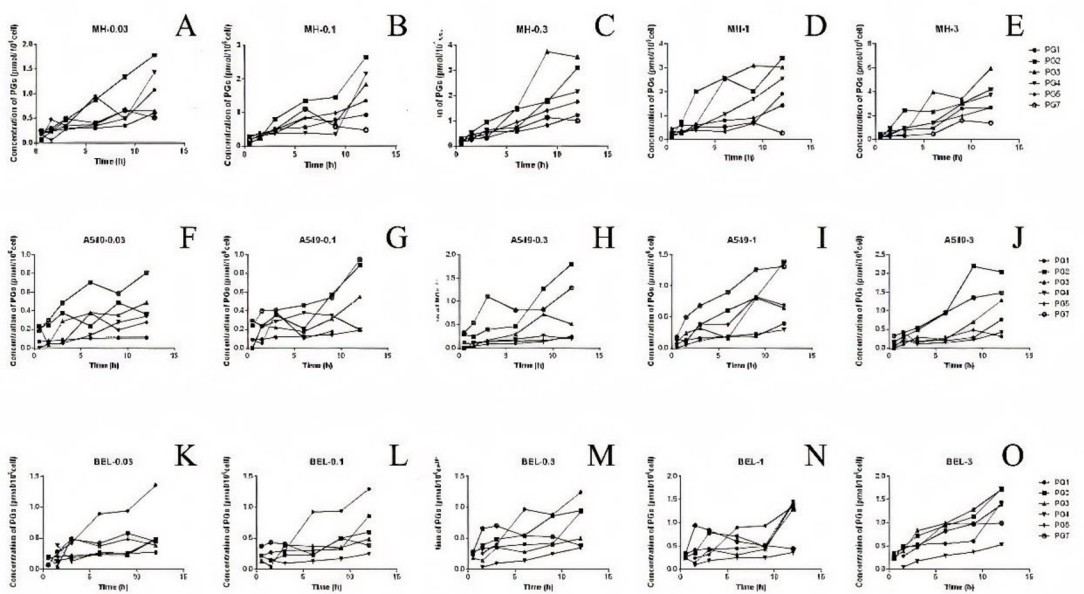

**Fig 4.** Curves of intracellular MTX-PG concentration over time in different cell lines treated with gradient methotrexate (A, B, C, D and E are the change curves of intracellular MTX-PGs after MHCC97H is treated with 0.03, 0.1, 0.3, 1 and 3μM MTX, respectively; F, G, H, I and J are the change curves of intracellular MTX-PGs after A549 was treated with 0.03, 0.1, 0.3, 1 and 3μM MTX, respectively; K, L, M, N and O are the change curves of intracellular MTX-PGs after BEL-7402 was treated with 0.03, 0.1, 0.3, 1 and 3μM MTX, respectively).

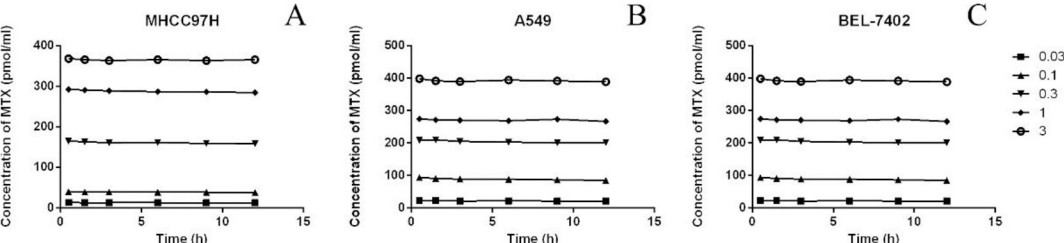

**Fig 5.** Curves of environmental methotrexate concentration over time in different cell lines treated with gradient MTX (A: the change process of MHCC87H environmental concentration under gradient MTX culture; B: the change process of A549 environmental concentration under gradient MTX culture; C: the change process of BEL-7402 environmental concentration under gradient MTX culture).

related studies have shown that this difference is related to the differential expression of MTX-related metabolic enzymes [10].

## 3.4 Correlation between the level of key enzymes and metabolic processes

**3.4.1 Determination of enzyme level in different cell lines.** The level of FPGS and GGH in cell samples, was quantified at various time points under gradient dose of MTX (0.03–3 μM) using the validated method. Student's t-test was applied for data analysis. In all three cell lines, intracellular abundance of both FPGS and GGH was relatively stable during 12-hour-long MTX exposure. There were significant differences in the abundance of enzymes in different cell lines. The abundance of FPGS in MHCC97H was higher than that in the other two groups, and the abundance of GGH in A549 was the highest (Fig 6).

The intracellular metabolism of drugs is affected by the abundance of the relevant enzymes. Different abundances of FPGS and GGH explain the difference in response to MTX in different tissues and organs. Studies [21] have shown that the expression of FPGS in B lymphocytes is three times that of T lymphocytes, resulting in greater sensitivity of MTX to B lymphocytes. Therefore, the expression level of intracellular MTX metabolizing enzyme affects the efficacy and side effects of MTX.

**3.4.2 Intracellular pharmacokinetic model fitting and parameter evaluation.** The obtained data of intracellular MTX-PG concentration and protein abundance was subjected to the established cytopharmacokinetics model, and the least square method was used to fit the principle. The fitting effect was as follows (Table 3).

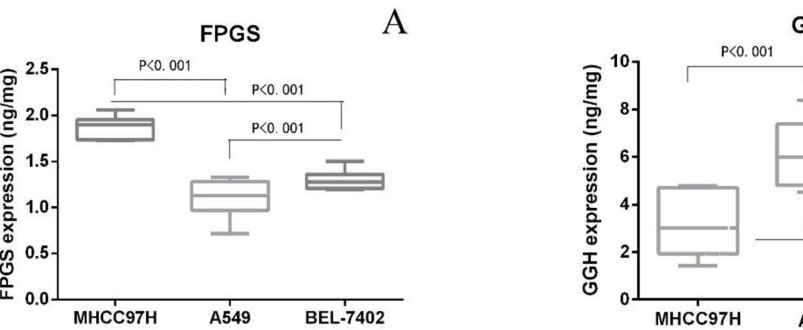

**Fig 6. Abundance of FPGS and GGH in different cell lines.**

**Table 3. Fitting results of methotrexate cytopharmacokinetics model.**

| Cell lines | $c_{MTX}$(μM) | $k_{in}$ | $k_{out}$ | $v_{max\text{-}FPGS}$ | $k_{GGH}$ | $R^2$ |
|---|---|---|---|---|---|---|
| MHCC97H | 0.03 | 0.35 | 0.61 | 3.1 | 4.5 | 0.96 |
| | 0.1 | 0.307 | 0.57 | 3.8 | 4.6 | 0.96 |
| | 0.3 | 0.39 | 0.55 | 4.3 | 1.06 | 0.67 |
| | 1 | 0.604 | 0.64 | 3.2 | 4.04 | 0.96 |
| | 3 | 0.532 | 0,79 | 5.5 | 2.97 | 0.94 |
| A549 | 0.03 | 0.19 | 1.04 | 0.19 | 14.25 | 0.81 |
| | 0.1 | 0.098 | 1.22 | 0.16 | 12.09 | 0.80 |
| | 0.3 | 0.24 | 0.32 | 0.49 | 14.94 | 0.99 |
| | 1 | 0.24 | 0.86 | 0.32 | 7.53 | 0.91 |
| | 3 | 0.44 | 1.73 | 0.42 | 6.72 | 0.87 |
| BEL-7402 | 0.03 | 0.49 | 1.8 | 0.34 | 9.41 | 0.95 |
| | 0.1 | 0.034 | 0.18 | 0.37 | 1.85 | 0.96 |
| | 0.3 | 0.046 | 0.13 | 0.38 | 0.88 | 0.98 |
| | 1 | 0.015 | 0.19 | 1.99 | 1.67 | 0.96 |
| | 3 | 0.27 | 0.44 | 1.23 | 1.24 | 0,88 |

* indicates unsuccessful fit

The results described above indicate that there is a rate-limiting step in the metabolism of intracellular MTX. This study divided the intracellular MTX kinetic process description into three parts, including the transport of MTX, the rate-limiting synthesis process of intracellular MTX-PG$_{2\text{-}7}$, and the hydrolysis of MTX-PG$_{2\text{-}7}$.

The transport of MTX: MTX can influx through active transport and passive diffusion, and efflux through active transport. Although the transport process depends on functional proteins, the transport rate is not limited. Moreover, the concentration of MTX in the environment is constant. The concentration of intracellular and extracellular MTX has a strong correlation and satisfies the first-order kinetic process.

The rate-limiting synthesis of intracellular MTX-PG$_{2\text{-}7}$: The results indicated that there is a significant rate-limiting step in the intracellular synthesis of MTX-PG$_{2\text{-}7}$. The enzyme FPGS that catalyzes this process is saturated. Vmax can be used to describe the rate of MTX-PG$_{2\text{-}7}$ synthesis when FPGS is saturated.

The hydrolysis of MTX-PG$_{2\text{-}7}$: The concentration of synthesized MTX-PG$_{2\text{-}7}$ was much lower than that of intracellular MTX concentration. Therefore, GGH-mediated MTX-PG$_{2\text{-}7}$ hydrolysis has no significant rate-limiting step, which can be described by the first-order kinetic equation.

The key pharmacokinetics parameters of MTX cells were obtained by model fitting. The effect of model fitting in different cell lines was good, and the intracellular metabolism of the drug was accurately described. The parameter fitting results were in the same range as the literature research data [20], suggesting proves that the model has good biological significance.

**3.4.3 Correlation analysis between the functional parameters of the model and protein expression.** Pearson correlation analysis was performed on the abundance of FPGS and GGH and the corresponding functional parameters Vmax and $k_{GGH}$. The abundance of FPGS was strongly correlated with $v_{max}$-FPGS (r = 0.854) (Fig 7), and the abundance of GGH was weakly correlated with $k_{GGH}$ (r = 0.614) (Fig 8).

The model parameter $v_{max}$-FPGS represents the maximum rate of intracellular MTX-PG generation when FPGS is saturated. The abundance of FPGS was strongly correlated with

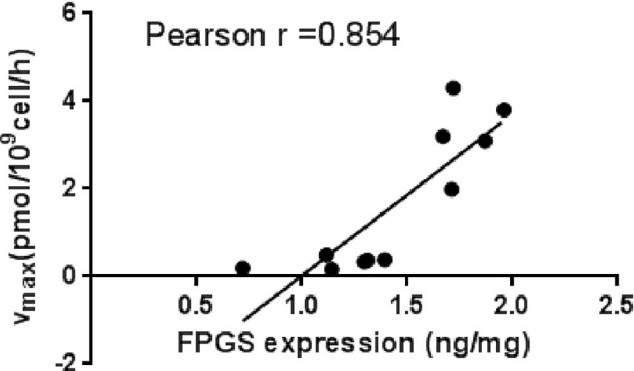

**Fig 7. The relationship of FPGS and $v_{max\text{-}FPGS}$.**

$v_{max}$-FPGS, indicating that the abundance of FPGS was the key factor determining the maximum rate of intracellular MTX. The parameter $k_{GGH}$ represents the first-order kinetic constant of GGH catalyzing the hydrolysis of MTX-PG$_{2\text{-}7}$. The correlation between the abundance of GGH and $k_{GGH}$ was relatively weak, indicating that the abundance of GGH dose not determine $k_{GGH}$. It was reported [22] that the activity of the same enzyme was not exactly the same in different cell lines. Therefore, $k_{GGH}$ may be affected by both enzyme abundance and unit enzyme activity. However, when FPGS is saturated, the differences in enzyme activity can be neglected, and the abundance of the enzyme limits the rate of MTX-PG production.

In summary, the abundance of FPGS determines the synthesis rate of MTX-PG in different cell lines. FPGS can be saturated with high dose of substrates, which limits the rate of intracellular MTX-PG synthesis. The abundance of GGH affects the hydrolysis rate of MTX-PG. The hydrolysis process can be also affected by the substrate concentration. Therefore, compared with FPGS, the effect of GGH on the intracellular concentration of MTX-PG is relatively small. In high-dose MTX treatment, FPGS is usually saturated and the intracellular MTX-PG synthesis rate is limited. The difference in plasma concentration between fast/slow infusion at the same dose has little effect on the synthesis rate of MTX-PG. During slow infusions, longer

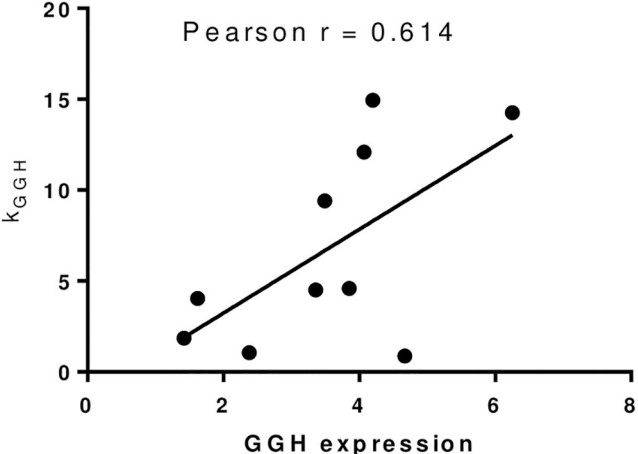

**Fig 8. The relationship of GGH and $k_{GGH}$.**

duration of exposure to MTX results in higher concentrations of MTX-PG and more serious side effects. The results suggest that the abundance of the intracellular metabolic enzyme FPGS can be used as an important indicator for ndividualized treatment with MTX. It can be used for predicting the accumulation of intracellular MTX-PG and drug efficacy.

In this study, the effects of FPGS and GGH on the intracellular metabolism of MTX were verified in vitro. Further studies are needed to validated this mechanism in vivo.

## 4 Conclusion

In this study the cytopharmacokinetic mechanism leading to more severe adverse reactions after slower MTX infusion at the same dose was determined in A549, BEL-7402 and MHCC97H cell lines. FPGS was saturated and the accumulation rate of MTX-PG was limited when excessive MTX was administered. The exposure time of MTX was prolonged under slow infusion, increasing the accumulation of MTXPG, which aggravated side effects. This study provides strategies for adjusting the MTX regimen and predicting efficacy/side effects. This study provides a new method for studying the intracellular metabolism of drugs.

## Author Contributions

**Data curation:** Le Zou.

**Formal analysis:** Le Zou.

**Investigation:** Jingyao Chen.

**Software:** Jingyao Chen.

**Writing – original draft:** Fang Tang.

**Writing – review & editing:** Fanqi Meng.

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
