## [Decision Letter · Decision Letter 0]

21 Feb 2024

PONE-D-24-04745Cellular pharmacokinetic mechanisms of methotrexate accumulation and its modulation by folylpolyglutamate synthetase and γ-glutamyl hydrolase in tumor cellsPLOS ONE

Dear Dr. Fanqi,

Thank you for submitting your manuscript to PLOS ONE. After careful consideration, we feel that it has merit but does not fully meet PLOS ONE’s publication criteria as it currently stands. Therefore, we invite you to submit a revised version of the manuscript that addresses the points raised during the review process.

We look forward to receiving your revised manuscript.

Kind regards,

Yawei Du

Academic Editor

PLOS ONE

Journal Requirements:

"This work was supported by Hunan Provincial Natural Science Foundation of China (S2017JJMSXM1571), Hunan Key Laboratory for Bioanalysis of Complex Matrix Samples (2017TP1037) and Fundamental Research Funds for the Central Universities of the Central South University (2019zzts759)"

4. We note that your Data Availability Statement is currently as follows: ""All relevant data are within the manuscript and its Supporting Information files.""

5. PLOS requires an ORCID iD for the corresponding author in Editorial Manager on papers submitted after December 6th, 2016. Please ensure that you have an ORCID iD and that it is validated in Editorial Manager. To do this, go to ‘Update my Information’ (in the upper left-hand corner of the main menu), and click on the Fetch/Validate link next to the ORCID field. This will take you to the ORCID site and allow you to create a new iD or authenticate a pre-existing iD in Editorial Manager. Please see the following video for instructions on linking an ORCID iD to your Editorial Manager account: https://www.youtube.com/watch?v=_xcclfuvtxQ.

Reviewers' comments:

Reviewer's Responses to Questions

**Comments to the Author**

1. Is the manuscript technically sound, and do the data support the conclusions?

Reviewer #1: Yes

Reviewer #2: Yes

2. Has the statistical analysis been performed appropriately and rigorously? 

Reviewer #1: Yes

Reviewer #2: Yes

3. Have the authors made all data underlying the findings in their manuscript fully available?

Reviewer #1: Yes

Reviewer #2: Yes

4. Is the manuscript presented in an intelligible fashion and written in standard English?

Reviewer #1: Yes

Reviewer #2: No

5. Review Comments to the Author

Reviewer #1: In this manuscript, the authors studied the metabolic process of methotrexate in cells based on the newly established UPLC -MS/MS absolute quantification method of FPGS. They established a cellular pharmacokinetic model of methotrexate and investigated the correlation between the abundance of key enzymes, FPGS and GGH, and corresponding functional parameters. Moreover, the authors elucidated the intrinsic mechanism of the phenomenon that slower infusion with more side effects. Overall, this article is informative while there still remains some problems.

1. Please note the consistency of the manuscript, there are only six curves in Figure 2 without MTX-PG6, while Table 1 has MTX-PG6, and the same problem appears in Figure 4.

2. In section 3.2, “However, it was found in this study that most of the accumulation concentrations of MTXPG2-6 after 24 hours of gradient methotrexate treatment had no correlation with the treatment methotrexate concentration”, is there any data to support this conclusion?

3. In section 3.3, “There are differences in the accumulation process of MTXPG between different cell lines, and related studies have shown that this difference is related to the differential expression of methotrexate-related metabolic enzymes”, please cite relevant references.

4. Please mark all abbreviations when they first appear in the text, such as “UPLC-MS”.

5. The quality of the figures in the manuscript requires improvement (such as Figure 2). Please provide clearer, high-resolution images and ensure that all texts are legible.

6. The authors may consider revising the language in the manuscript to improve the clarity and readability of the text.

Reviewer #2: The manuscript“Cellular pharmacokinetic mechanisms of methotrexate accumulation and its modulation by folylpolyglutamate synthetase and γ-glutamyl hydrolase in tumor cells” established UPLC-MS/MS absolute quantification method of FPGS and GGH. However, the manuscript is lack of novelty. The whole experimental design only stays at the cellular level. I recommend to accept this manuscript after miajor revision.

(1)The English writing should be carefully edited by a native speaker of a proofread professional.

(2)Fig 1 should be redrawn to make this picture more artistic.

(3)Some language and grammar errors still should be carefully checked through the manuscript. In abstract, the format and language need to be modified.

(4)The A,B,C font in fig2 and fig3 is inconsistent.

(5)Some in vivo experiments need to be supplemented.

6. PLOS authors have the option to publish the peer review history of their article (what does this mean?). If published, this will include your full peer review and any attached files.

Reviewer #1: No

Reviewer #2: No

---

## [Author Response · Author response to Decision Letter 0]

26 Mar 2024

Dear the editor and reviewers: 

Thank you very much for giving us an opportunity to revise our manuscript PONE-D-24-04745) titled Cellular pharmacokinetic mechanisms of methotrexate accumulation and its modulation by folylpolyglutamate synthetase and γ-glutamyl hydrolase in tumor cells. We appreciate the positive and constructive comments and suggestions from the editor and reviewers, which significantly improved the quality of our manuscript. We have revised the manuscript accordingly. Both the comments and the responses are listed below point by point.

Editorial office comments:

Response: Thank you very much for the good advice, We have checked throughout the manuscript and revised.

Response: Thank you very much for your reminding. We have ensured the code is shared in a way that follows best practice and facilitates.

"This work was supported by Hunan Provincial Natural Science Foundation of China (S2017JJMSXM1571), Hunan Key Laboratory for Bioanalysis of Complex Matrix Samples (2017TP1037) and Fundamental Research Funds for the Central Universities of the Central South University (2019zzts759)"

Response: Sorry for this error. We have removed the fundings-related text from the manuscript. And we ensure that the authors received no specific fundings for this word.

4. We note that your Data Availability Statement is currently as follows: ""All relevant data are within the manuscript and its Supporting Information files.""

Response: Thank you very much for your reminding. We have ensured the all relevant date are within the manuscript and its supporting information files.

5. PLOS requires an ORCID iD for the corresponding author in Editorial Manager on papers submitted after December 6th, 2016. Please ensure that you have an ORCID iD and that it is validated in Editorial Manager. To do this, go to ‘Update my Information’ (in the upper left-hand corner of the main menu), and click on the Fetch/Validate link next to the ORCID field. This will take you to the ORCID site and allow you to create a new iD or authenticate a pre-existing iD in Editorial Manager. Please see the following video for instructions on linking an ORCID iD to your Editorial Manager account: https://www.youtube.com/watch?v=_xcclfuvtxQ.

Response: Thank you very much for your reminding. We have ensured that we have an ORCID in Editorial Manager.

Reviewer #1: In this manuscript, the authors studied the metabolic process of methotrexate in cells based on the newly established UPLC -MS/MS absolute quantification method of FPGS. They established a cellular pharmacokinetic model of methotrexate and investigated the correlation between the abundance of key enzymes, FPGS and GGH, and corresponding functional parameters. Moreover, the authors elucidated the intrinsic mechanism of the phenomenon that slower infusion with more side effects. Overall, this article is informative while there still remains some problems.

1. Please note the consistency of the manuscript, there are only six curves in Figure 2 without MTX-PG6, while Table 1 has MTX-PG6, and the same problem appears in Figure 4.

Response: Thank you very much for your constructive suggestions. In our experiment. The concentration of MTX-PG6 is much smaller than other metabolites, which failed to map effectively with the concentration of other metabolites, and this date has been used in other articles.

2. In section 3.2, “However, it was found in this study that most of the accumulation concentrations of MTXPG2-6 after 24 hours of gradient methotrexate treatment had no correlation with the treatment methotrexate concentration”, is there any data to support this conclusion?

Response: Thank you very much for the good advice. We added the relevant reference to support this conclusion. And the details have been provided in section 3.2.

3. In section 3.3, “There are differences in the accumulation process of MTXPG between different cell lines, and related studies have shown that this difference is related to the differential expression of methotrexate-related metabolic enzymes”, please cite relevant references.

Response: Thank you very much for the good advice. We have cited the relevant reference to support “There are differences in the accumulation process of MTX-PG between different cell lines, and related studies have shown that this difference is related to the differential expression of MTX-related metabolic enzymes”, and

details have been provided in section 3.3.

4. Please mark all abbreviations when they first appear in the text, such as “UPLC-MS”.

Response: Sorry for these errors. We have checked throughout the manuscript and revised.

5. The quality of the figures in the manuscript requires improvement (such as Figure 2). Please provide clearer, high-resolution images and ensure that all texts are legible.

Response: Sorry for these errors. We have checked and revised figures.

6. The authors may consider revising the language in the manuscript to improve the clarity and readability of the text.

Response: Thank you very much for the good advice. Our manuscript has been edited by a native English-speaking expert to ensure its English is good enough for publication. And the details have been provided in manuscript.

Reviewer #2: The manuscript“Cellular pharmacokinetic mechanisms of methotrexate accumulation and its modulation by folylpolyglutamate synthetase and γ-glutamyl hydrolase in tumor cells” established UPLC-MS/MS absolute quantification method of FPGS and GGH. However, the manuscript is lack of novelty. The whole experimental design only stays at the cellular level. I recommend to accept this manuscript after miajor revision.

(1)The English writing should be carefully edited by a native speaker of a proofread professional.

Response: Thank you very much for the good advice. Buy your suggestion, our manuscript has been edited by a native English-speaking expert to ensure its English is good enough for publication. And the details have been provided in manuscript.

(2)Fig 1 should be redrawn to make this picture more artistic.

Response: Thank you very much for the good advice. We used the colored picture to revise our manuscript, and have specific response to the dear reviewers.

Fig 1 Cellular and external MTX transport process and intracellular methotrexate metabolism process (Vmax is the saturation rate when intracellular MTX-PG1 is converted to PG2-7; kGGH is the kinetic constant)

(3)Some language and grammar errors still should be carefully checked through the manuscript. In abstract, the format and language need to be modified. 

Response: Thank you very much for the good advice. Buy your suggestion, our manuscript has been edited by a native English-speaking expert to ensure its English is good enough for publication. And the details have been provided in manuscript, and have specific response to the dear reviewers.

(4)The A,B,C font in fig2 and fig3 is inconsistent.

Response: Thank you very much for the good advice. We have checked and revised the figures.

(5)Some in vivo experiments need to be supplemented.

Response: Thank you very much for the nice suggestion. We deeply agree that results from in vivo experiments would enhance the depth of our research，thereby elevating the quality of our paper. However, in the present study, we mainly focus on firmed that the abundance of FPGS is the decisive factor limiting the synthesis rate of MTX-PG, and we think the present experiments may not be optimal, but should be sufficient to draw a conclusion that “ the abundance of FPGS is the decisive factor limiting the synthesis rate of MTX-PG”. We will conduct in vivo experiments as you suggested in the future work.

---

## [Decision Letter · Decision Letter 1]

10 Apr 2024

Cellular pharmacokinetic of methotrexate and its modulation by folylpolyglutamate synthetase and γ-glutamyl hydrolase in tumor cells

PONE-D-24-04745R1

Dear Dr. Meng,

We're pleased to inform you that your manuscript has been judged scientifically suitable for publication and will be formally accepted for publication once it meets all outstanding technical requirements.

Within one week, you'll receive an e-mail detailing the required amendments. When these have been addressed, you'll receive a formal acceptance letter and your manuscript will be scheduled for publication.

An invoice will be generated when your article is formally accepted. Please note, if your institution has a publishing partnership with PLOS and your article meets the relevant criteria, all or part of your publication costs will be covered. Please make sure your user information is up-to-date by logging into Editorial Manager at Editorial Manager® and clicking the "Update My Information" link at the top of the page. If you have any questions relating to publication charges, please contact our Author Billing department directly at authorbilling@plos.org.

If your institution or institutions have a press office, please notify them about your upcoming paper to help maximize its impact. If they'll be preparing press materials, please inform our press team as soon as possible -- no later than 48 hours after receiving the formal acceptance. Your manuscript will remain under strict press embargo until 2 pm Eastern Time on the date of publication. For more information, please contact onepress@plos.org.

Kind regards,

Yawei Du

Academic Editor

PLOS ONE

Additional Editor Comments (optional):

Reviewers' comments:

Reviewer's Responses to Questions

**Comments to the Author**

1. If the authors have adequately addressed your comments raised in a previous round of review and you feel that this manuscript is now acceptable for publication, you may indicate that here to bypass the “Comments to the Author” section, enter your conflict of interest statement in the “Confidential to Editor” section, and submit your "Accept" recommendation.

Reviewer #1: All comments have been addressed

Reviewer #2: (No Response)

2. Is the manuscript technically sound, and do the data support the conclusions?

Reviewer #1: Yes

Reviewer #2: Yes

3. Has the statistical analysis been performed appropriately and rigorously? 

Reviewer #1: Yes

Reviewer #2: Yes

4. Have the authors made all data underlying the findings in their manuscript fully available?

Reviewer #1: Yes

Reviewer #2: Yes

5. Is the manuscript presented in an intelligible fashion and written in standard English?

Reviewer #1: Yes

Reviewer #2: Yes

6. Review Comments to the Author

Reviewer #1: (No Response)

Reviewer #2: (No Response)

7. PLOS authors have the option to publish the peer review history of their article (what does this mean?). If published, this will include your full peer review and any attached files.

Reviewer #1: No

Reviewer #2: No

---

## [Editor Report · Acceptance letter]

24 Apr 2024

PONE-D-24-04745R1 

PLOS ONE

Dear Dr. Meng, 

I'm pleased to inform you that your manuscript has been deemed suitable for publication in PLOS ONE. Congratulations! Your manuscript is now being handed over to our production team.

Kind regards, 

on behalf of

Dr. Yawei Du 

Academic Editor

PLOS ONE